# Associations between self-efficacy, distress and anxiety in cancer patient-relative dyads visiting psychosocial cancer support services: Using actor-partner interdependence modelling

**Solveigh P. Lingens**●*, **Florian Schulz, Isabell Müller, Holger Schulz, Christiane Bleich**

Department of Medical Psychology, University Medical Centre Hamburg-Eppendorf, Hamburg, Germany

* s.lingens@uke.de

## Abstract

**Data Availability Statement:** The data file is available from https://osf.io/qfbuc/.

### Background

Patients with cancer and their relatives often suffer from psychosocial burdens following a cancer diagnosis. Psychosocial cancer support services offer support for cancer patients and their relatives. Only a few studies have focused on associations of psychological factors within patient-relative dyads. This study aims to assess associations between the patients' or relatives' self-efficacy and their levels of distress and anxiety who seek help together at psychosocial cancer support centres.

### Methods

Participants were recruited at two psychosocial cancer support centres in a major city in Germany. Patients with cancer and their relatives seeking support together received the questionnaire before their first support session. Self-efficacy was assessed with the Pearlin sense of mastery scale, distress with the distress thermometer and anxiety with the General Anxiety Disorder questionnaire (GAD-7). For the analysis, the actor-partner interdependence model was applied.

### Results

The data analysis was based on 41 patient-relative dyads (patients: 39% women, mean age 53.5; relatives: 66% women, mean age 52.16). A significant actor effect from self-efficacy to distress was found for patients ($r = -0.47$) but not for relatives ($r = -0.15$). Partner effects from self-efficacy to distress were not significant ($r = -0.03$, $r = -0.001$). The actor effect from self-efficacy to anxiety for patients ($r = -0.61$) as well as relatives was significant ($r = -0.62$), whereas the partner effect was significant for patients ($r = 0.16$) but not for relatives ($r = -0.46$).

**Funding:** This study was funded by the Hamburger Krebsgesellschaft e.V. (Cancer Society Hamburg). HS and CB received the funding. The funders had no role in study design, data collection and analysis, decision to publish, or preparation of the manuscript.

**Competing interests:** The authors have declared that no competing interests exist.

## Conclusion

The results suggest that patients' and relatives' self-efficacy is associated with their distress and anxiety. Partner effects were visible for patients' self-efficacy and relatives' anxiety. These findings suggest that self-efficacy is an important factor for the psychological well-being of patients and relatives and that it may additionally be associated with the partners' well-being. Longitudinal research with larger samples is needed to support the findings.

## Introduction

The life expectancy and physical health of patients diagnosed with cancer has increased as medical treatments have improved over the last few years [1]. Due to the higher survival rates, interventions that focus on improving the mental health of patients with cancer have gained importance [2, 3]. A cancer diagnosis and its treatment can affect the psychological well-being of a patient. Most patients suffer from psychological distress resulting from the diagnosis, treatment or other consequences. Furthermore, patients diagnosed with cancer face different life changes, for example, occupational and financial hazards [4].

As the patients' duration of stay in the hospital has decreased [5] and cancer treatment has been progressively translocated into outpatient settings [6] the responsibilities of caregivers have grown [7]. Not only do relatives of patients with cancer have to cope with the consequences of the illness, but they also often function as informal (unpaid) caregivers [8]. Research has shown that relatives show similar distress levels as cancer patients [9]. Moreover, cancer-related anxiety seems to be higher in informal caregivers than in patients, whereas patients show greater symptoms of depression [10, 11]. Patients and their relatives both have to deal with the consequences of a cancer diagnosis on a practical and emotional level. Relatives are crucial in assisting the patients in making key decisions regarding the cancer treatment and in providing emotional support [12]. Couple research has shown that the couples' reactions towards the diagnosis and their adjustment are highly interdependent [12]. Studies examining this interdependence in dyads of a patient and the partner concluded that patients' and their relatives' reactions to stress deriving from advanced or incurable cancer are interdependent [11, 12].

One major psychological factor associated with the psychological well-being of cancer patients as well as their relatives is self-efficacy [13, 14]. In the context of cancer, self-efficacy can be defined as the degree to which patients and relatives have trust in their capacity to manage the consequences of a cancer diagnosis [15]. It may also function as a resource that can moderate negative outcomes [16]. First of all, cancer-related self-efficacy can influence patients' and caregivers' mental and physical health. A study on the interdependence of self-efficacy in couples coping with advanced cancer revealed that a person's self-efficacy is not influenced by the other person's self-efficacy [17]. However other variables as mental and physical health were found to be interdependent in patients and caregivers [17]. Even if one person's self-efficacy does not affect the other person's self-efficacy, it can influence the partner's psychological health [17]. Other findings also suggest that caregivers showing high self-efficacy can improve patients' mental well-being [18].

Distress and anxiety are two of the most common psychological symptoms experienced by cancer patients and their relatives [19]. Research suggests that self-efficacy may be interrelated with distress as well as anxiety. In a study on the influence of dyadic distress in advanced cancer greater distress was associated with lower individual and family-related self-efficacy [13].

Beyond that, a meta-analysis by Chirico et al. revealed an inversed correlation between cancer-related self-efficacy and distress [15]. Studies on the relationship between anxiety and self-efficacy in advanced cancer patients have found a significant link between self-efficacy and levels of anxiety [20]. But only a few studies have focused on the patient-relative dyad instead of considering patients and their relatives as separate units [17].

To address the psychosocial burdens of a cancer diagnosis easy-accessible, low-threshold psychosocial support for patients with cancer and their relatives is needed. Psychosocial cancer support services offer psychosocial support for patients with cancer and are often available for relatives of patients with cancer [21]. A number of psychosocial cancer support services offer support for family members together. Although patients and relatives suffer from psychological distress quite similarly, the concerns that they wish to address in a support session may differ. Differences in general support needs between patients and relatives were found in a cross-sectional study examining patient-reported needs in outpatient psychosocial support centres [22]. Whereas patients were rather concerned about legal advice, relatives mostly requested psychological support. Studies on caregivers' support needs revealed that caregivers have unmet informational, psychological and emotional concerns [9, 11, 12]. Although differences of patients and relatives seeking advice at psychosocial cancer support services were identified, patients and relatives seeking help together have not yet been considered in research [22].

To the best of our knowledge there are no studies that have investigated the associations between self-efficacy, distress and anxiety in patient-relative dyads visiting psychosocial cancer support services. Examining how the self-efficacy of couples that seek advice at psychosocial cancer support services is associated with distress and anxiety may indicate whether combined support could be beneficial. Furthermore, insights into the dynamics between patients and their relatives may help to adjust the couple's support to improve the benefits of the support services for both, patients and relatives. Therefore, the objective of the study is to provide first insights into the dynamics of patients and relatives seeking help at two psychosocial cancer support centres in Germany. Hence, the aim is to determine whether the patient's or relative's level of self-efficacy is associated with their levels of distress or anxiety and/or with the accompanying person's level of distress or anxiety.

## Methods

### Study design

The present study is a cross-sectional study of patients with cancer and their relatives seeking advice at two psychosocial cancer support centres in a major city in Germany. The following findings result from an analysis of baseline measurements of an ongoing larger quasi-experimental prospective study to evaluate the effectiveness of psychosocial cancer support centres [23].

### Setting

Patients with cancer and their relatives who sought advice in one of the two psychosocial cancer support centres in Hamburg, Germany from December 2018 –March 2020 were included. Before the first support session, patients and relatives received oral and written information on the aims and procedure of the study and were asked to participate. If they agreed to take part in the study, they gave their written consent. Subsequently, they received the baseline questionnaire and were given sufficient time to complete it before the support session started. The support was administered from psycho-oncologists, social workers and doctors depending on the concern of the clients. Psychological, social or legal concerns were addressed. The complete procedure of the full study is explained in more detail elsewhere [23].

## Participants

Cancer patients and relatives of cancer patients were eligible to participate if they had contacted one of the psychosocial cancer support centres for the first time during the recruitment phase, had sufficient German language skills to complete the questionnaire, had no severe physical, mental or cognitive constraints and were over 18 years old. For the dyadic analysis, only participants who came as a pair of two were included, independent of their relationship (e.g., partner, sibling, parent).

## Ethical considerations

The study was approved by the local psychological ethical committee of the Centre of Psychosocial Medicine (LPEK) at the University Medical Centre Hamburg-Eppendorf on August 22, 2018 (No: LPEK-007) in Hamburg, Germany.

## Instruments

**Pearlin sense of mastery scale.** The Pearlin Sense of Mastery Scale was used to assess self-efficacy. It combines aspects of perceived self-efficacy and the construct of the internal locus of control. It was originally developed by Pearlin and Schooler [24]. The original version includes 7 items such as "I have little control over the things that happen to me". For this study, the short version of four items was used [25]. All items are negatively worded and require reverse coding before scoring. The response options were offered on a four-point Likert scale. The scores range from 4 to 16, with higher scores indicating a greater level of sense of mastery. The questionnaire meets the test quality criteria [26] and correlates with several scales measuring well-being and depression [27].

**Distress thermometer.** The distress thermometer is sensitive for measuring distress in cancer patients [28]. It consists of a general distress score from 0 to 10, where 10 indicates high distress. For values above the cut-off (> 5), it is assumed that there is a need for psychological support. Besides, the distress measure encompasses a detailed list of 35 problems that provide more information on possible reasons for the level of distress. The list groups the problems in categories such as practical, family, emotional, physical and spiritual concerns. Participants can indicate whether any of the problems apply to them by ticking "Yes" or "No". The German version of the measure has been validated and is often used in clinical practice as a screening tool [29].

**General Anxiety Disorder questionnaire (GAD-7).** The patient health questionnaire (PHQ) was originally developed and tested in a large study including over 6,000 patients. Because the original measure included 27 items assessing a variety of health aspects, it was shortened and separated into individual questionnaires such as the General Anxiety Disorder questionnaire (GAD-7) amongst others [30, 31] to identify generalized anxiety disorder as well as assess symptom severity of generalized anxiety disorder. Items of the GAD-7 are scored from 0 (not at all) to 3 (nearly every day), with higher scores demonstrating more serious symptoms of anxiety. It is widely used to screen for anxiety, especially social anxiety, panic and PTSD [32].

**Confounding variables.** Since the sample is relatively small, only gender was included in the analysis. Gender is associated with levels of anxiety and distress and also with seeking support, where women tend to rather seek support and show higher levels of anxiety and distress compared to men [12, 33].

## Statistical methods

The actor–partner interdependence model (APIM) [34, 35] represents a well-established statistical tool for analyzing the interdependence of certain variables in dyads [36]. In the present

study the APIM regression for distinguishable dyads was used to examine the effects of a person's self-efficacy on his or her distress/anxiety (i.e., the actor effect) and on the distress/anxiety of his or her partner (i.e., the partner effect). Gender was included as a covariate to control for possible confounding influences. The statistical analysis was performed with the online app called "APIM_SEM" developed by Stas et al. [37]. It uses the program lavaan, an R-package, for structural equation modelling with maximum likelihood estimation to fit the APIM [37]. Two separate APIM models were computed to assess the effect of self-efficacy on distress and self-efficacy on anxiety. The power was calculated based on 41 available dyads in the sample. With the alpha level set at 0.05 and an effect size of partial $r = 0.4$ for the actor effect and partial $r = 0.25$ for partner effects, the power is estimated to be 75% and 34% respectively [38]. Although small effect sizes are not likely to be detected with the available sample, the analysis was performed, and the power was taken into account as a limitation in the interpretation of the results.

## Results

### Characteristics of the sample

Of the 450 cancer patients and relatives of cancer patients, who sought advice in one of the two psychosocial cancer support centres in a major city in Germany from December 2018 –March 2020, 42 dyads of cancer patients and relatives sought advice and were thus eligible for this study. Of those, 41 patient-relative dyads completed the baseline measures. The average age of patients was 53.5 years (*s.d.* = 14.0) and the average age of relatives was 53.2 years (*s.d.* = 13.5). Most patients were men (61%) and most relatives were women (66%). Twenty-six dyads were married couples (63%), seven were couples in an intimate relationship (17%), six were family members (15%), and for two the information about their relationship was missing (5%). Of all participants, 71% had at least one child. Concerning the financial situations, 68% of the relatives and 32% of the patients were employed (Table 1).

For patients, the most common type of cancer was lung cancer (22%) followed by breast cancer (17%), prostate cancer (7%), oral cancer (7%), stomach cancer (7%), skin cancer (2%), and other cancer types (38%). 34% reported metastases. More than half of the patients received chemotherapy (51%) and half of the patients were undergoing surgery (49%). Other treatments included antibody/immunotherapy (17%) and antihormonal therapy (12%). The average time since the first diagnosis was 22.5 months (*s.d.* = 50.5).

The types of support requested were social and legal support (78%), psychological support (44%) and medical support (15%). For relatives, social and legal support (78%) and psychological support (45%) were equally important compared to patients, whereas medical support (28%) was slightly more relevant.

Relatives reported significantly higher self-efficacy and lower distress compared to patients, whereas levels of anxiety were similar. The differences between patients and relatives are presented in Table 1.

### Associations between self-efficacy and distress

The APIM revealed a statistically significant actor effect from self-efficacy to distress for patients (Fig 1). Patients who reported higher self-efficacy showed lower levels of distress. For relatives, the actor effect was not statistically significant. However, a small effect size was detected ($r = 0.15$), which indicates that the effect may be clinically relevant but the analysis lacks sufficient power to detect it. The difference between the actor effects of patients and relatives was not found to be statistically significant. The overall actor effect from self-efficacy to distress remained statistically significant.

**Table 1. Differences between patients and relatives.**

| | Patients (n = 41) | Relatives (n = 41) | t-Test / χ² | Effect size |
|---|---|---|---|---|
| **Age** [*mean (s.d.)*] | 53.50 (14.09) | 53.16 (13.51) | $t = 0.108$ | $d = 0.03$ |
| **Sex** [%] | | | $\chi^2 = 5.917^*$ | $\Phi = 0.27$ |
| Female | 39 | 66 | | |
| Male | 61 | 34 | | |
| **Migrant background** [%] | | | $\chi^2 = 0.78$ | $\Phi = 0.10$ |
| Yes | 13 | 15 | | |
| No | 87 | 85 | | |
| **Family status** [%] | | | $\chi^2 = 2.684$ | $\Phi = 0.18$ |
| Single | 10 | 5 | | |
| Married | 71 | 73 | | |
| In a relationship | 15 | 15 | | |
| Separated | 2 | 0 | | |
| Divorced | 2 | 7 | | |
| **Living situation** [%] | | | $\chi^2 = 4.333$ | $\Phi = 0.23$ |
| Alone | 0 | 3 | | |
| Shared flat | 5 | 5 | | |
| With partner | 54 | 48 | | |
| With partner and children | 37 | 43 | | |
| With children | 0 | 2 | | |
| With parents | 5 | 0 | | |
| **Education** [%] | | | $\chi^2 = 2.156$ | $\Phi = 0.16$ |
| General secondary school | 15 | 10 | | |
| Intermediate secondary school | 32 | 29 | | |
| Grammar school/high school | 22 | 22 | | |
| Apprenticeship | 15 | 10 | | |
| University/College | 17 | 29 | | |
| **Occupation** [%] | | | $\chi^2 = 0.436$ | $\Phi = 0.07$ |
| Full-time | 57 | 50 | | |
| Part-time | 22 | 24 | | |
| Studying | 3 | 3 | | |
| Unemployed | 5 | 8 | | |
| Pension | 14 | 16 | | |
| **Source of income** [%] | | | $\chi^2 = 19.477^{**}$ | $\Phi = 0.49$ |
| Self-employed | 0 | 8 | | |
| Employed | 32 | 68 | | |
| Continued pay | 3 | 0 | | |
| Sick pay | 42 | 8 | | |
| Unemployment benefit 1 | 3 | 0 | | |
| Unemployment benefit 2 | 5 | 3 | | |
| Pension | 16 | 13 | | |
| **Type of support** [%] | | | $\chi^2 = 1.259$ | $\Phi = 0.12$ |
| Social/legal | 78 | 78 | | |
| Psychological | 44 | 45 | | |
| Medical | 15 | 28 | | |
| **Self-efficacy** [*mean (s.d.)*] | 10.29 (3.02) | 11.85 (2.74) | $t = -2.396^*$ | $d = 0.54$ |
| **Anxiety** [*mean (s.d.)*] | 8.87 (5.20) | 7.95 (5.33) | $t = 0.734$ | $d = 0.18$ |

(*Continued*)

**Table 1.** (Continued)

| | Patients (n = 41) | Relatives (n = 41) | *t*-Test / $\chi^2$ | Effect size |
|---|---|---|---|---|
| **Distress** [*mean (s.d.)*] | 7.28 (1.55) | 6.08 (2.41) | *t* = 2.619* | *d* = 0.66 |

**Notes**.
*$p < 0.05$
**$p < 0.01$
***$p < 0.001$

The partner effect from patients' self-efficacy to relatives' distress was not statistically significant (Table 2). Furthermore, there was no statistically significant partner effect from relatives' self-efficacy to patients' distress. This indicates that the reported self-efficacy of patients and relatives was not associated with their partners' levels of distress. The difference between the partner effects of patients and relatives was not found to be statistically significant. Overall, the partner effect from self-efficacy to distress was not statistically significant.

## Associations between self-efficacy and anxiety

The actor effect from self-efficacy to anxiety was statistically significant for patients and relatives (Fig 2). Patients and relatives who reported higher self-efficacy showed lower levels of anxiety. There was no statistically significant difference between the actor effects of patients and relatives. The overall actor effect from self-efficacy to anxiety was statistically significant.

The partner effect from patients' self-efficacy to relatives' anxiety was statistically significant. Relatives of patients with higher self-efficacy showed lower levels of anxiety. In contrast, the partner effect from relatives' self-efficacy to patients' anxiety was not statistically significant. However, a small effect size was detected (*r* = 0.16), which suggests a lack of power to detect the effect. This difference between the partner effects of patients and relatives was found to be statistically significant. The overall partner effect from self-efficacy to anxiety was not statistically significant.

## Associations with gender

The analysis showed a significant association between the patients' gender and the patients' levels of distress (*t* = -1.097, *p* = .013, 95% CI [-1.964, -0.229]), with an overall standardized effect of *β* = -0.349. Male patients showed lower levels of distress compared to female patients.

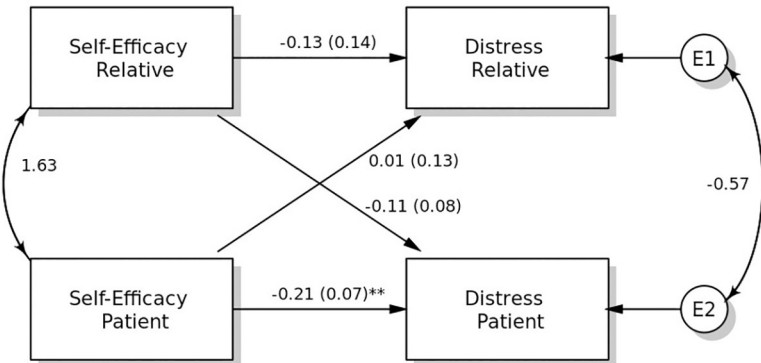

**Fig 1. The standard model for self-efficacy on distress.**

**Table 2. Results of the actor-partner interdependence model.**

|  | Effect |  | $\chi^2$ | b [s] | b [o] | (se) | t | CI | Partial r |
|---|---|---|---|---|---|---|---|---|---|
| **Anxiety** |  |  | 22.57* |  |  |  |  |  |  |
| Patients | Actor |  |  | -0.59 | -0.33 | 0.22 | -1.01*** | [-1.46, -0.59] | -0.61 |
|  | Partner |  |  | 0.07 | 0.07 | 0.25 | 0.13 | [-0.37, 0.63] | 0.16 |
| Relatives | Actor |  |  | -0.53 | -0.56 | 0.22 | -1.03*** | [-1.47, -0.58] | -0.62 |
|  | Partner |  |  | -0.37 | -0.36 | 0.20 | -0.65** | [-1.05, -0.26] | -0.46 |
| **Distress** |  |  | 35.09*** |  |  |  |  |  |  |
| Patients | Actor |  |  | -0.40 | -0.47 | 0.07 | -0.21** | [-0.34, -0.07] | -0.47 |
|  | Partner |  |  | -0.20 | -0.03 | 0.08 | -0.11 | [-0.41, 0.15] | -0.03 |
| Relatives | Actor |  |  | -0.15 | -0.19 | 0.14 | -0.13 | [-0.41, 0.15] | -0.15 |
|  | Partner |  |  | 0.01 | 0.01 | 0.13 | 0.01 | [-0.24, 0.26] | -0.001 |

Notes. APIM: Actor-Partner Interdependence Model. $\chi^2$ value represents the test of distinguishability. If the test is significant the dyad members are distinguishable. Betas are provided as overall standard deviation across all persons [o] and separately for patients and relatives [s]. Beta [o] should be considered if betas are to be compared across dyad members.

*$p < 0.05$

**$p < 0.01$

***$p < 0.001$.

The gender of relatives was not associated with their own levels of distress ($t = 0.047$, $p = .953$, 95% CI [-1.509, 1.603]), with an overall standardized effect of $\beta = 0.009$. The gender of patients and relatives showed also no statistically significant association with their levels of anxiety (patients: $t = -2.081$, $p = .133$, 95% CI [-4.797, 0.636]; relatives: $t = -0.712$, $p = .574$, 95% CI [-3.191, 1.768]). The overall standardized effect of gender on the patients' levels of anxiety was $\beta = -0.197$ and on the relatives' levels of anxiety $\beta = -0.064$.

## Discussion

This is the first study to examine the associations of self-efficacy, distress and anxiety in patient-relative dyads that seek support at psychosocial cancer support centres together.

In this study, most patients were male and most relatives were female, which resembles the typical population in studies with patients and caregivers [39, 40]. Since women are generally more likely to seek psychological help than men are [33], women may be more likely to feel the need to accompany their ill partner compared to men. Whereas, if female patients with

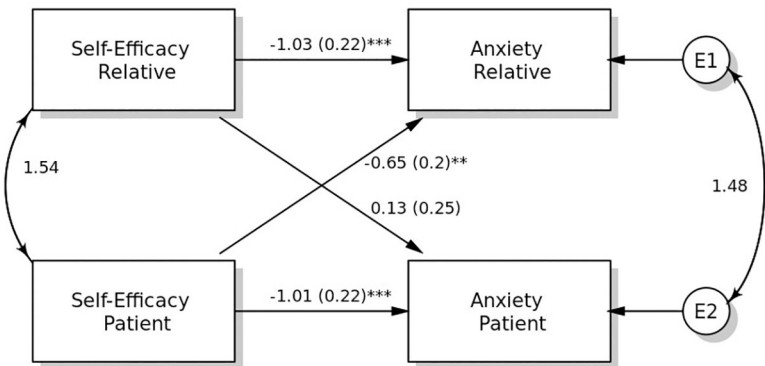

**Fig 2. The standard model for self-efficacy on anxiety.**

cancer seek help, their male partner may not feel the need for help. Considering these findings for this study, only for patients, gender was associated with distress, where men had lower levels of distress compared to women. In line with the results of other studies, women with cancer tend to show higher levels of distress than men [12]. Interestingly, the relatives' gender was not associated with distress. Since relatives had relatively low levels of distress compared to patients, one explanation may be that female and male relatives may have served as support and did not have any concerns for themselves. The difference in gender between patients and relatives did not influence the results. This suggests that gender may not have any notable influence on actor or partner effects. In other words, the association between self-efficacy and the own and partners' well-being is independent of gender. The significant difference in the source of income, where twice as many relatives were employed compared to patients, is likely due to patients having to report sick more often.

Concerning the main analysis, we expected to find a significant association between the patients' and relatives' self-efficacy and their distress. A significant actor effect from self-efficacy to distress was found for patients but not for relatives. These results comply with findings of other studies, that found an association between self-efficacy and distress in patients with cancer [41] and specifically in female cancer patients [42, 43]. For relatives, the evidence is inconsistent with our findings. One study found an association between perceived control and life satisfaction and depression among elderly caregivers [44], whereas another study found an association only among female caregivers of patients with cancer [45]. However, the low power of our study to detect small effects needs to be considered when interpreting these results and maybe the reason for the association to be non-significant.

Second, regarding the partner effects, we expected that the patients' and relatives' self-efficacy would be interrelated with their partner's level of distress. Surprisingly, our results did not reveal significant partner effects from self-efficacy to distress, neither for patients nor relatives. These findings disagree with the findings of one study that investigated associations between self-efficacy and psychological distress in patients with lung cancer and their caregivers [14]. The level of self-efficacy of caregivers was associated with levels of patients' distress [14]. Generally, these findings suggest that the level of self-efficacy can serve as an important indicator for the level of psychological distress of patients. In the clinical setting psychosocial interventions could aim to improve self-efficacy and coping skills of both patients and relatives to improve their well-being as has been previously established [46, 47].

Third, as hypothesized, we found a significant actor effect from self-efficacy to anxiety for patients as well as relatives. These findings comply with existing research on the association between anxiety and self-efficacy in advanced cancer patients [14, 20]. Furthermore, we found a significant inverse association between patients' self-efficacy and their relatives' levels of anxiety, which complies with the results of similar research [14, 17].

Finally, the partner effect from self-efficacy to anxiety was not found to be significant for relatives, which may be due to low power to detect small effect sizes in this study. However, these results still suggest that an improvement of self-efficacy in either patients or their relatives is not just associated with their level of anxiety but also with the anxiety of their partner, where the association between patients' self-efficacy and their relatives' anxiety seems to be stronger than vice versa. One explanation may be that if the patients feel more in control and more self-efficient, their relative feels more reassured and has less to worry about the patient.

Regarding the clinical implications of the results of the dyadic analysis, it may be valuable for psycho-oncologist or social worker who provide psychosocial support to focus on the improvement of self-efficacy in both patients and their relatives, to yield a maximal improvement of their psychological well-being. Professionals may want to assess similarities and differences in distress and anxiety symptoms between patients and their relatives and tailor the

support accordingly. To consider partner effects during psychosocial support it may be recommendable to generally make support available for patients and as well as their relatives. Moreover, oncologists who are usually in close contact with their patients should inform about psychosocial support offers, especially for their relatives. Since self-efficacy is a relevant factor for the improvement of psychological well-being, the focus should be on improving the feeling of control for patients as well as their relatives. If relatives are involved in the process of coping with the disease, they can be supported with defining their role and regaining control, which could positively affect their and the patients' self-efficacy and well-being. In other support domains, for example, HIV counselling, research has shown that counselling seems to be more effective if it is received as a couple [48]. Longitudinal studies with patient-relative dyads are needed to further understand the dynamics of psychosocial interventions within a couple.

### Study limitations

There were limitations to this study. First, the number of dyads that were included in the analysis was relatively small. In the original study on the effectiveness of cancer support services, only 14% of the included participants sought help as a couple. Thus, the number of dyads eligible for inclusion was limited. To detect small effect sizes a larger sample would have been necessary especially for partner effects. However, significant effects were still detected and hence meaningful conclusions were derived. Nevertheless, the generalizability should be done with caution since the sample is rather small and includes only people who have decided to seek help. Hence, dyadic dynamics may be different for patients and their relatives who do not seek help.

Furthermore, the APIM model does not allow the interpretation of the results in terms of causality. Self-efficacy could lead to increased improvement of distress and anxiety or vice versa. However, some studies have established the causal impact of self-efficacy on distress [46, 47]. Nevertheless, the APIM is an appropriate method to explore dyadic associations as it is theoretically coherent as other studies have found similar associations.

A third limitation might be, that the quality of the relationships was not assessed and could have added valuable knowledge to improve the interpretations of the results. Seeking help together in the first place may be an indicator that the quality of the relationship is rather good. Although the information was not available on the reasons why some couples were seeking advice together and others did not, some explanations are more reasonable than others. The relatives' motivations may have resulted from the wish to show their support to their partner or to raise questions concerning relationship topics. As these are only assumptions, the motivation and reasoning behind the decision to come as a couple may have provided more qualitative insights into the dynamics of the couple.

Another limitation is that confounding variables that have not been controlled for may have influenced the results. The age of the participants or severity of the cancer diagnosis may have had an impact on the associations between self-efficacy, distress and anxiety within a patient-relative dyad. Influences of other confounders should be considered for further studies with a higher number of participants.

Despite these limitations, the results of this study provide insights into dyadic dynamics between patients and relatives, who seek out psychosocial support, which will allow providers to tailor interventions specific to the needs of couples.

## Conclusion

This is the first study to focus on associations between self-efficacy, distress and anxiety within cancer patient-relative dyads seeking psychosocial support. Despite the small sample, the

results of the study confirm that patients' and relatives' self-efficacy is associated with their distress and anxiety. Partner effects were visible for patients' self-efficacy and relatives' anxiety. A small, but non-significant effect size was found for an association between relatives' self-efficacy and patients' anxiety. These findings suggest that self-efficacy is an important factor for the psychological well-being of patients and relatives and that it may additionally be associated with the partners' well-being. Therefore, one implication for the clinical setting may be an expansion of psychosocial cancer support offers for couples and relatives. Further qualitative and longitudinal research with larger samples would be useful to scrutinize the complexity of dyadic effects between cancer patients and their relatives within the context of psychosocial support.

## Author Contributions

**Conceptualization:** Solveigh P. Lingens, Christiane Bleich.

**Data curation:** Florian Schulz, Isabell Müller.

**Formal analysis:** Solveigh P. Lingens, Florian Schulz, Isabell Müller.

**Funding acquisition:** Holger Schulz, Christiane Bleich.

**Methodology:** Solveigh P. Lingens.

**Project administration:** Solveigh P. Lingens.

**Supervision:** Solveigh P. Lingens, Holger Schulz, Christiane Bleich.

**Writing – original draft:** Solveigh P. Lingens, Florian Schulz, Isabell Müller.

**Writing – review & editing:** Solveigh P. Lingens, Christiane Bleich.

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
