## [Decision Letter · Decision Letter 0]

23 Jun 2021

PONE-D-21-12900

Associations between self-efficacy, distress and anxiety in cancer patient-relative dyads visiting psychosocial cancer support services: Using actor-partner interdependence modelling

PLOS ONE

Dear Dr. Lingens,

Thank you for submitting your manuscript to PLOS ONE. After careful consideration, we feel that it has merit but does not fully meet PLOS ONE’s publication criteria as it currently stands. Therefore, we invite you to submit a revised version of the manuscript that addresses the points raised during the review process.

Please see comments below. 

We look forward to receiving your revised manuscript.

Kind regards,

Andrew Soundy

Academic Editor

PLOS ONE

Journal Requirements:

Additional Editor Comments:

Introduction

can you double check the context of the last paragraph and statement that says little is known about... line 116. As a reader I would want to know if you are aware of any work? most importantly that has done something similar to your study, if so name it and identify limitations, if not say to the best of your knowledge none. But also be mindful of other design types which may bring something to the table. Any contextual studies added to give insight to the reader can be taken through to the discussion also. You conclusion says only a few studies have looked at associations line 380 - so please make it clear what is known, not known and what further work is needed and making it easy for the reader to understand your choices.

Methods

can you make sure you use sub-heading according to STROBE https://www.strobe-statement.org/index.php?id=available-checklists

for instance take requirements for the area of setting they ask for periods of recruitment, exposure, follow-up, they ask for attempts to address sources o bias, sampling and sample size, also identification of how confounding variables are controlled for in the analysis - you identify gender? wasn't their many others? e.g., mental health, level of physical activity, impact of COVID-19 (sampling period means people could be affected by it) I notice your discussion has some e.g., quality of social support network but I am not sure you cover all possibilities etc. Can you justify your choice and provide limitations if any further are identified as a result of this.

Reviewers' comments:

Reviewer's Responses to Questions

**Comments to the Author**

1. Is the manuscript technically sound, and do the data support the conclusions?

Reviewer #1: Yes

2. Has the statistical analysis been performed appropriately and rigorously? 

Reviewer #1: Yes

3. Have the authors made all data underlying the findings in their manuscript fully available?

Reviewer #1: Yes

4. Is the manuscript presented in an intelligible fashion and written in standard English?

Reviewer #1: Yes

5. Review Comments to the Author

Reviewer #1: The authors present a cross-sectional study evaluating self-efficacy, distress and anxiety in dyads of cancer patients and their relatives being advised at psychosocial cancer support centers. This analysis is part of a lager study evaluating efficacy of cancer support centers. The authors were able to analyze a total 42 dyads which is a high number respecting the difficulties to include dyads in prospective studies.

The underlying study concerns on an issue of very high clinical impact as all three factors are frequent and relevant in daily practice in oncologic care of cancer patients and their relatives, but less is known about interaction between this three dimensions and between the two dyad partners.

The manuscript is written interestingly and comprises all relevant previous literature. The study is performed on a high methodological standards and all results are presented in an adequate and comprehensive manner.

In my opinion, this manuscript is suitable for publication in Plos One, but there is one aspect that could improve the overall value of the manuscript and should be revised previously:

The authors should extend their implications for clinical practice. It strengthens the importance of psychosocial cancer support services, but I think there could be more clinical implications. What can oncologist, psycho-oncologist social workers in cancer support services learn from this study for their daily practice? What should they now do to improve their support for cancer patients, their relatives and both. What can they advise them?

6. PLOS authors have the option to publish the peer review history of their article (what does this mean?). If published, this will include your full peer review and any attached files.

Reviewer #1: No

---

## [Author Response · Author response to Decision Letter 0]

8 Jul 2021

Journal Requirements:

Reply: We have doubled-checked the format of our manuscript and cannot find any deviations from the PLOS ONE style requirements.

Reply: We have made our data available on the public repository OSF. The link to see and download the data is.

https://osf.io/qfbuc/?view_only=2fa4df7d78d746b9bc41e98a5395440b

The file shows the original minimal data to replicate our analysis. For an analysis with the APIM_SEM app, it needs to be converted into a dyad format.

Reply: We have checked every reference. None of the papers we have cited have been retracted. 

Additional Editor Comments:

Introduction

can you double check the context of the last paragraph and statement that says little is known about... line 116. As a reader I would want to know if you are aware of any work? most importantly that has done something similar to your study, if so name it and identify limitations, if not say to the best of your knowledge none. But also be mindful of other design types which may bring something to the table. Any contextual studies added to give insight to the reader can be taken through to the discussion also. You conclusion says only a few studies have looked at associations line 380 - so please make it clear what is known, not known and what further work is needed and making it easy for the reader to understand your choices.

Reply: Thank you for the remark. We have named a number of articles that have done similar studies in different contexts (situations outside the support context, specific cancer types, male vs female dayds etc.) throughout the introduction and the discussion. However, there are indeed no studies investigating associations between self-efficacy, distress and anxiety in dyads of patients and relatives in the context of seeking psychosocial support. We understand our phrasing in line 116 and 380 may be misleading so we have changed the phrasing in line 116 and deleted the sentence in line 380.

Methods

can you make sure you use sub-heading according to STROBE https://www.strobe-statement.org/index.php?id=available-checklists

for instance take requirements for the area of setting they ask for periods of recruitment, exposure, follow-up, they ask for attempts to address sources of bias, sampling and sample size, also identification of how confounding variables are controlled for in the analysis - you identify gender? wasn't their many others? e.g., mental health, level of physical activity, impact of COVID-19 (sampling period means people could be affected by it) I notice your discussion has some e.g., quality of social support network but I am not sure you cover all possibilities etc. Can you justify your choice and provide limitations if any further are identified as a result of this.

Reply: We have edited the methods section according to the STROBE criteria. The sub-headings variables, data sources/measurement and quantitative variables were grouped together under the sub-heading instruments. We have only controlled for gender as our sample size is relatively small and adding more confounders to the analysis would have resulted in a loss of power. Gender was the strongest confounder. We have added this explanation in the limitation section. We have included all participants before the outbreak of COVID-19 in Germany. Furthermore, we believe the mental health of the participants is already sufficiently explained by the variables anxiety and distress. Physical activity was not assessed in the study. Since participants had to be able to travel to the support service facilities, we assume a certain level of physical activity needs to be given. We discuss biases in the limitation section and therefore feel it may be redundant to name them also in the method section. We have added some information on the exposure but did not elaborate on the follow-up since this is a cross-sectional study and may lead to confusion. We have cited the study protocol of the larger study. 

Reviewer #1: 

The authors present a cross-sectional study evaluating self-efficacy, distress and anxiety in dyads of cancer patients and their relatives being advised at psychosocial cancer support centers. This analysis is part of a lager study evaluating efficacy of cancer support centers. The authors were able to analyze a total 42 dyads which is a high number respecting the difficulties to include dyads in prospective studies.

The underlying study concerns on an issue of very high clinical impact as all three factors are frequent and relevant in daily practice in oncologic care of cancer patients and their relatives, but less is known about interaction between this three dimensions and between the two dyad partners.

The manuscript is written interestingly and comprises all relevant previous literature. The study is performed on a high methodological standards and all results are presented in an adequate and comprehensive manner.

In my opinion, this manuscript is suitable for publication in Plos One, but there is one aspect that could improve the overall value of the manuscript and should be revised previously:

The authors should extend their implications for clinical practice. It strengthens the importance of psychosocial cancer support services, but I think there could be more clinical implications. What can oncologist, psycho-oncologist social workers in cancer support services learn from this study for their daily practice? What should they now do to improve their support for cancer patients, their relatives and both. What can they advise them?

Reply: Thank you for reading our manuscript and thank you for the remark. You are right, there are more implications for clinical practice. We have added some more possible implications (see line 343-360).

---

## [Decision Letter · Decision Letter 1]

14 Jul 2021

Associations between self-efficacy, distress and anxiety in cancer patient-relative dyads visiting psychosocial cancer support services: Using actor-partner interdependence modelling

PONE-D-21-12900R1

Dear Dr. Lingens,

We’re pleased to inform you that your manuscript has been judged scientifically suitable for publication and will be formally accepted for publication once it meets all outstanding technical requirements.

Kind regards,

Andrew Soundy

Academic Editor

PLOS ONE

Additional Editor Comments (optional):

Reviewers' comments:

Reviewer's Responses to Questions

**Comments to the Author**

1. If the authors have adequately addressed your comments raised in a previous round of review and you feel that this manuscript is now acceptable for publication, you may indicate that here to bypass the “Comments to the Author” section, enter your conflict of interest statement in the “Confidential to Editor” section, and submit your "Accept" recommendation.

Reviewer #1: All comments have been addressed

2. Is the manuscript technically sound, and do the data support the conclusions?

Reviewer #1: Yes

3. Has the statistical analysis been performed appropriately and rigorously? 

Reviewer #1: Yes

4. Have the authors made all data underlying the findings in their manuscript fully available?

Reviewer #1: Yes

5. Is the manuscript presented in an intelligible fashion and written in standard English?

Reviewer #1: Yes

6. Review Comments to the Author

Reviewer #1: In my opinion, after this comprenhsive revision, this manuscript is now suitable for publication in POLOS one.

7. PLOS authors have the option to publish the peer review history of their article (what does this mean?). If published, this will include your full peer review and any attached files.

Reviewer #1: No

---

## [Editor Report · Acceptance letter]

9 Sep 2021

PONE-D-21-12900R1 

Associations between self-efficacy, distress and anxiety in cancer patient-relative dyads visiting psychosocial cancer support services: Using actor-partner interdependence modelling 

Dear Dr. Lingens:

I'm pleased to inform you that your manuscript has been deemed suitable for publication in PLOS ONE. Congratulations! Your manuscript is now with our production department. 

Kind regards, 

on behalf of

Dr. Andrew Soundy 

Academic Editor

PLOS ONE